# An Experimental Investigation of Hardwoods Harvested in Croatian Forests for the Production of Glued Laminated Timber

**DOI:** 10.3390/ma16051843

**Published:** 2023-02-23

**Authors:** Ivana Uzelac Glavinić, Ivica Boko, Jelena Lovrić Vranković, Neno Torić, Mario Abramović

**Affiliations:** 1Faculty of Civil Engineering, Architecture and Geodesy, University of Split, 21000 Split, Croatia; 2Drvene Konstrukcije D.o.o, 33522 Voćin, Croatia

**Keywords:** glued laminated timber, hardwoods, European hornbeam, Turkey oak, maple, surface treatment

## Abstract

The aim of this study was to assess the potential of hardwoods harvested in Croatian forests for the production of glued laminated timber (glulam), mainly of those species for which there is no published performance assessment. Nine sets of glulam beams were produced: three sets using lamellas from European hornbeam, three sets from Turkey oak, and three sets from maple. Each set was characterized by a different hardwood species and surface preparation method. The surface preparation methods included planing, planing followed by sanding with fine grit, and planing followed by sanding with coarse grit. The experimental investigations included shear tests of the glue lines in dry conditions and bending tests of the glulam beams. The shear tests showed satisfactory performance of the glue lines for the Turkey oak and European hornbeam, but not for the maple. The results of the bending tests showed superior bending strength of the European hornbeam compared to the Turkey oak and maple. Planing followed by rough sanding of the lamellas was shown to have a significant influence on the bending strength and stiffness of the glulam from Turkey oak.

## 1. Introduction

Timber is no longer considered an outdated building material with limited mechanical properties. Today, some skyscrapers are made almost entirely out of engineered wood products (EWPs) [1,2,3], and the use of timber is promoted due to its environmental benefits [4,5,6]. EWPs such as glulam and cross-laminated timber (CLT) allow us to choose sustainable materials and cut and grade lamellas and glue them onto a construction product with controlled mechanical properties and dimensions. A key part of this process is ensuring the integrity of the glue line. According to the European Standards for glulam made of coniferous (softwood) species or poplar [7], the bonding strength of glue lines should be either verified by a delamination test or a shear test. The shear test is to be performed on service classes 1 and 2.

Glulam production is dominated by softwoods. Softwoods have a faster growth rate, good workability, and a simpler production process (primarily gluing). In recent years, the continuous felling of softwoods and climatic changes that favor drought-tolerant species have led to an increase in the stock of hardwoods in Europe [8,9]. Currently, 46% of European forests are coniferous (softwood) forests, 37% are deciduous (hardwood) forests, and 17% are mixed. The proportion of hardwoods increases from north to south. Hardwoods cover about 15% of the forest area in northern countries such as Sweden and Finland, over 40% in Central Europe, and more than 60% in southeastern countries. In Croatia, deciduous trees cover up to 82% of the total forest area [9]. Due to environmental factors, the share of deciduous forests is expected to further increase. As a result, the use of hardwoods as a building material at the expense of softwoods is strongly promoted [10,11,12].

A series of studies on the application of hardwoods for the production of glulam was recently conducted in Europe [13,14,15,16,17,18]. The species that were investigated include *Fagus sylvatica* L. (European beech), *Quercus petraea* L. (Sessile oak), *Quercus robur* L. (Pedunculate oak), *Castanea sativa* L. (Chestnut), and *Fraxinus excelsior* L. (Ash). It was shown that glulam made of softwoods (i.e., fir and spruce), in comparison to glulam made of hardwood, had higher density and tensile strength, which leads to superior mechanical properties [19,20]. A stumbling block to a wider use for glulam is securing the integrity of the glue lines, i.e., a lack of related standards. The difficulty in creating a standard is primarily due to the diversity in hardwood species; namely, hardwoods have a wide density range, and some species are incompatible with certain types of adhesives for reasons that are not always known. Experimental data for each species are needed in order to establish production processes that will satisfy the delamination or shear tests. In 2021, the European Organisation for Technical Assessment (EOTA) published the European Assessment Document (EAD) for glulam made of solid hardwood, which is limited to certain hardwood species [21]. For species not included in the EAD, technical approval may be requested at the level of an individual state based on experimental results. The experimental results must demonstrate sufficient bonding strength of the glue lines, and manufacturers may vary a number of parameters until a satisfactory production process is obtained.

In the case of glulam made of softwood to be used in service classes 1 and 2, the bonding strength of glue lines is verified by the shear test [7], and the results must not exceed the thresholds given for shear strength and wood failure percentage (WFP). In the case of glulam made of hardwood, for species included in the EAD [21], the delamination and shear tests of the lamellas at their mid-section should also be performed. The reason for the latter is that bonding quality can be estimated with the bond shear strength and wood shear strength ratio [22]. The study [22] also states that the ratio is persistent regardless of the production process or hardwood species, so it is expected to be implemented in future European standards for glulam made of hardwood. Moreover, the EAD [21] provides two options for the bond shear strength test: in dry and in wet conditions.

The aim of this paper is to present experimental investigations on glulam made from hardwood species that are present in Croatian forests but are not included in the EAD [21]. The introduction of such species to the market requires experimental results, which are currently lacking. The production processes for the glulam in this study included three different surface treatments of the lamellas, i.e., three different sets for each hardwood species, and they are presented in detail. The bond line integrity was tested via shear tests of the bond lines and lamellas in dry conditions. Bending tests on the glulam beams with lengths up to 4 m, i.e., without finger joints, were performed. A discussion on the suitability of selected species for the production of glulam is also included.

## 2. Materials and Methods

Three different hardwood species were used for the production of glulam beams: *Carpinus betulus* L. (European hornbeam), *Quercus cerris* L. (Turkey oak), and *Acer campestre* L. (Maple). All three species are available in Croatian forests and are mostly used as energy sources. European hornbeam is the most widespread of them and accounts for 8.4% of the total wood stock in Croatia, followed by Turkey oak with 1.8%, and Maple with less than 1% [23]. Figure 1 shows the distribution map of the presented species within the European area [24]. In this study, all timber was supplied from Virovitica-Podravina County, Croatia, at a 200 m height above sea level. All timber was from mixed and natural forests near Papuk mountain. The trees from European hornbeam, Turkey oak, and maple were 77 to 79 years old, 75 to 78 years old, and 82–84 years old, respectively. Timber was processed at the glulam manufacturing plant owned by Drvene konstrukcije Ltd. Since no adhesive has been approved for European hornbeam, the manufacturer used the existing technology with few modifications in the gluing process, as described below. Modifications were necessary since the existing technologies have been adjusted for spruce and fir. Higher densities of hardwood compared to spruce were paired with lower porosity, causing penetration of the adhesive to occur at a slower rate. The morphological structure of European hornbeam and maple is diffuse-porous, with sparse pores being more widely spaced for maple. On the other hand, Turkey oak is ring-porous wood, with wider pores in the earlywood compared to the pores in latewood. In addition to the choice of the adhesive and gluing process, the surface preparation of the lamellas is also known to have an influence on the quality of the bond line. This influence is particularly noticeable in moisture-related performance, but studies investigating this effect are rare [25,26]. In this study, for the production of glulam, three different surface treatments of the lamellas (planing, sanding with grit 60, and sanding with grit 80) were performed in order to study their effects on the properties of glulam.

### 2.1. Production of Glulam Beams

The wood was conditioned at the manufacturing plant at a room temperature of 20 °C and relative humidity of 65%. Lamellas were cut to the required length, and local defects such as knots were discarded. All lamellas were processed on a timber planning machine (Figure 2a). Prior to gluing, the moisture content of each lamella was measured with an electronic wood moisture meter (Gann Hydromette HT 65). The measured moisture content of the lamellas was between 8 and 15%, with the maximum difference in moisture for the lamellas in each beam not surpassing 2.5%. The mean density for maple was 620 kg/m^3^, for European hornbeam, it was 790 kg/m^3^, and for Turkey oak, it was 745 kg/m^3^.

For each hardwood species, one set of lamellas was planed up to a thickness of 2 cm, the second set was planed and then sanded with grit size 60 to the required thickness of 2 cm (rough sanding), and the third set was planed and then sanded with grit size 80 to the required thickness of 2 cm (fine sanding). Lamellas were glued with melamine–urea–formaldehyde adhesive (Prefere 4535) and hardener (Prefere 5035) from Dynea, at a ratio of 100:25. The adhesive was applied manually with 400 g/m2 of mixer and at a room temperature of 20 °C, in accordance with the technical sheet [27]. In consultation with the manufacturer of the adhesive, for the application of this adhesive onto hardwood species, the pot life was assumed to be 30 min. The assembly time, i.e., the time between the application of the adhesive and the application of pressure, consisted of an open assembly time, which lasted up to 5 min, and a closed assembly time, which lasted from 5 to 15 min. Afterwards, a cramping pressure of 0.8–1.2 N/mm^2^ was applied (Figure 2b). Finally, the glulam beams were planed on all sides in order to obtain the required dimensions of 60 × 80 × 1700 mm. Due to the total length of 1700 mm, no finger joints were produced in the glulam beams.

### 2.2. Relative Bond Shear Strength Tests

Relative bond shear strength tests in dry conditions were carried out in accordance with Annex A of the EAD [21], as shown in Figure 3a. Samples with a length of 50 mm were cut out from a complete glulam cross-section. The shear areas of each sample had sizes of 2500 mm^2^. The values measured were bond shear strength, wood shear strength, and wood failure percentage (WFP). The tests were carried out in a force-controlled regime using a universal testing machine with a constant cross-head displacement rate, as shown in Figure 3b.

Test pieces were placed between the jaws of a testing machine with the glue line oriented parallel to the loading direction. The bond shear strength $f_{v,b}$ of each bond line was calculated according to [7] as follows:(1)$$f_{v,b}=k\frac{F_{u}}{A}$$
where k=0.78+0.044t, *t* is the thickness of the sample, *F*_u_ is the failure load [N], and *A* is the sheared area [mm^2^]. The WFP is defined as the percentage of wood failure area in relation to the total sheared area. The WFP was estimated visually for every bond line directly after mechanical testing. Furthermore, the wood shear strength values of the lamellas loaded at their mid-plane ($f_{v,w}$) were also determined.

The relative mean bond shear strength is defined as the ratio of the mean value of the bond shear strength $f_{v,b,mean}$ and the wood shear strength $f_{v,w,mean}$:(2)$$rel\;f_{v,b,mean}=\frac{f_{v,b,mean}}{f_{v,w,mean}}$$

Likewise, the relative 5% quantile bond shear strength is defined as the ratio of the 5% quantile of the bond shear strength $f_{v,b,05}$ and the 5% quantile of the wood shear strength $f_{v,w,05}$:(3)$$rel\;f_{v,b,05}=\frac{f_{v,b,05}}{f_{v,w,05}}$$

### 2.3. Bending Test Setup

Bending tests were performed on the glulam beams in accordance with EN 408 [28]. The tests were conducted in an air-conditioned environment at the Structures laboratory of the Faculty of Civil Engineering, Architecture and Geodesy, Split. A moisture content of 8 ± 2% was measured for each glulam beam before it was tested. The geometry of the four-point bending test performed in this study is shown in Figure 4. The beams were simply supported with a span length of around 18 times the depth of the cross-section (*h*), and loaded symmetrically at two points at a distance of around 6x*h* from the supports.

Bending tests were conducted on an Automax Multitest machine, as shown in Figure 5. The load was applied with two symmetric loading cylinders via a rigid beam fixed under the cell. The constant loading-head movement of 8 mm/min was determined from preliminary tests in order to reach the maximum load *F*_max_ within a time interval of 300 ± 120 s. Displacements were measured continuously with displacement transducers (LVDTs) connected to an HBM QuantumX data acquisition system and recorded using software catmanEasy 5.3.1.27. The test procedure was divided into several phases. First, the beam was loaded until it reached 10% of the *F*_max_ and kept constant for 50 s. The test continued afterwards until 40% of the *F*_max_ was reached in order to measure the displacements and strain components needed for the local and global modulus of elasticity. Afterwards, the beam was unloaded until 10% of the *F*_max_ was reached, and the LVDTs were disassembled to avoid damage. Finally, the beam was loaded until failure and the *F*_max_ was recorded.

An Automax Multitest machine recorded the load and displacement of the rigid beam, while an HBM data acquisition system recorded the displacements of the rigid beam and displacements for the local and global modulus of elasticity (MOE). Specially composed aluminum bars with a distance of 5 × *h* and LVDTs positioned at the middle of the beam were used for measuring the relative strain required for the calculation of the local MOE in case of bending. In accordance with [28], the local MOE was determined from the load/displacement record between 0.1 *F*_max_ ($F_{1}$) and 0.4 *F*_max_ ($F_{2}$) using the following expression:(4)$$E_{m,l}=\frac{al_{1}^{2}\left(F_{2}-F_{1}\right)}{16I\left(w_{2}-w_{1}\right)}$$
where ($w_{2}-w_{1}$) is an increment of displacement at the neutral axis corresponding to $F_{2}\;\mathrm{and}\;F_{1}$, a is the distance between the loading position and the nearest support, $l_{1}$ is the gauge length (i.e., 5 × *h*), and *I* is the second moment of area. In order to calculate the global MOE in bending, the displacement *w* was measured at the bottom of the mid-section of the beam. Likewise, the load/displacement record between $F_{1}$ and $F_{2}$ was used to calculate the global MOE using the following equation:(5)$$E_{m,g}=\frac{3al^{2}-4a^{3}}{2bh^{3}\left(2\frac{w_{2}-w_{1}}{F_{2}-F_{1}}\right)}$$
where *l* is the length of the span in bending, and *b* and *h* are the width and depth of the specimen cross-section. It should be noted that the full expression for the global MOE given in [28] includes the shear modulus *G*. In this study, *G* was assumed to have an infinite value, which resulted in the slight modification of the expression. The bending strength $f_{m}$ was calculated for each glulam beam using the following equation:(6)$$f_{m}=\frac{aF_{max}}{2W}\;$$
where *W* is the section modulus.

## 3. Results and Discussion

### 3.1. Relative Bond Shear Strength Tests

EN 14080 [7] for softwoods requires the shear strength of each glue line to be at least 6 N/mm^2^, and the required minimum for the WFP is related to the shear strength. For hardwoods, however, it has been shown that the relative bond shear strength is a much better indicator of the bond line quality than the bond shear strength [22]. The requirements given in the EAD [21] for the shear test of the glue lines in dry conditions include the relative 5% quantile of bond shear strength $rel\;f_{v,b,05}\geq 0.9$ and the 10% quantile of the wood fiber percentage $WFP_{10}\geq 0.5$. Moreover, performance assessment (i.e., factory production control) requires the relative mean bond shear strength $rel\;f_{v,b,mean}\geq 0.9$ and mean wood fiber percentage $WFP_{mean}\geq 0.8$. Table 1 summarizes the relative bond shear strength and wood failure percentage, taking into account the different surface treatments for each hardwood species. It can be seen that for the glulam made from maple, the relative 5% quantile of bond shear strength and 10% quantile of wood fiber percentage did not meet the requirements of the EAD [21] in any surface treatments, i.e., $rel\;f_{v,b,05}$ was less than the required 0.9, and $WFP_{10}$ was less than 50%. The Turkey oak showed the highest values for the relative 5% quantile of bond shear strength and 10% quantile of wood fiber percentage by exceeding the required values for all surface treatments. The European hornbeam surpassed the requirements for the 5% quantile of bond shear strength and 10% quantile of wood fiber percentage (i.e., for initial type testing) with the exception of rough sanding (S60). The typical failure modes are shown in Figure 6.

It should be noted that larger sample sizes should be used to confirm these conclusions. The results shown here suggest that the application of melamine–urea adhesive (Prefere 4535) for the production of glulam from Turkey oak and European hornbeam can provide satisfying bonding quality. Further research is needed to better evaluate the influence of rough surface treatment (S60) on the bonding quality of European hornbeam. The gluing process described here is not recommended for the production of glulam from maple.

### 3.2. Bending Tests

For each tested glulam beam, the bending strength and the local and global MOE were calculated according to Equations (4)–(6) given in EN 408 [28]. The results of the bending tests of 55 glulam beams are summarized in Table 2. For each hardwood species column, P refers to glulam with planed lamellas, and S80 and S60 are glulam with planed and sanded lamellas with grit sizes of 80 and 60, respectively. The mean bending strength $f_{m,g,mean}$, the mean global modulus of elasticity $E_{m,g,mean}$, the local modulus of elasticity $E_{m,l,mean},$ and the standard deviation *SD* were calculated according to EN 14358 [29].

The results indicate superior bending strength for the hornbeam glulam compared to the Turkey oak and maple glulam. For the planed surfaces, the mean bending strength values of the hornbeam glulam were 46% and 64% higher than those of the maple and Turkey oak glulam, respectively. The higher bending strength of the hornbeam glulam was accompanied by higher bending stiffness, i.e., 55% and 49% higher than that of the Turkey oak and maple glulam, respectively. *SD_fm,g_* indicates the need for a greater number of samples for planed surfaces. Figure 7 presents a box plot of the bending strength, with the horizontal line within the box representing the median, and the horizontal lines outside of the box meaning the minimum and maximum values. Sanding resulted in higher bending strengths compared to planing, with 10 %, 11%, and 34% increases for the maple, Turkey oak, and hornbeam, respectively. Due to the small sample size, the influence of the surface preparation on the bending strength should be confirmed on a larger sample size. Moreover, the results for the global and local MOE summarized in Table 2 show clear increases in the bending stiffness for Turkey oak and rough sanding of the lamellas (S60) compared to just planing. The increases in the mean MOE were 35% and 43% for the local and global MOE, respectively.

The load–deflection data obtained from the bending tests are shown in Figure 8. A typical failure mode occurred in the outermost lamella in the tension zone, with a crack propagating through the lamella. As explained in Section 2, the LVDTs that measured the displacement at the bottom of the mid-section were disassembled before the loading force reached 40% of the F_max_ in order to avoid damage. Therefore, the deflection shown in Figure 8 corresponds to the deflection measured at the points where the force was inserted (see Figure 4), which was a few millimeters smaller than the deflection at the bottom of the mid-section. As can be seen in Figure 8, the load–deflection record corresponding to the maple and European hornbeam confirms similar bending behavior among different surface treatments. Figure 8b, i.e., the load–deflection record corresponding to the Turkey oak, clearly supports the above-mentioned discussion on higher bending strength and stiffness for the rough sanding of the lamellas (S60) compared to just planing (P) or fine sanding (S80). Higher bending strength is indicated by the higher load forces for rough sanding, while higher bending stiffness is indicated by the higher slopes of the deflection curves at the starting point of the graph.

The obtained bending strength and global and local MOE for each glulam beam were subjected to analysis of variance (one-way ANOVA) at a 95% confidence level (*p* ≤ 0.05) in order to evaluate the effects of different surface treatments of the lamellas on the previously mentioned properties. The results of the analysis indicate no statistically significant impact of different surface treatments of the lamellas on the bending strength and stiffness of glulam beams made of European hornbeam or maple. On the other hand, the ANOVA showed significant differences among the means of the bending strength and MOE of the glulam beams made of Turkey oak. Since the *p*-value of the F-test was less than 0.05, as can be seen in Table 3, there were significant differences among the means of the three groups regarding the different surface treatments.

Accordingly, Duncan’s Multiple Range Test (DMRT) was used post-hoc at a 95% confidence level (*p* ≤ 0.05) to determine which means of the bending strength and MOE of the glulam beams made of Turkey oak were significantly different from the others. Comparisons among the means were performed and compared with the least significant range. As can be seen in Table 4, significant differences were confirmed for the Turkey oak with planing and rough sanding of the lamellas (S60), compared to just planing (P) or fine sanding (S80) of the lamellas.

This study focused on glulam beams up to 4 m long without any finger joints. For these lengths, the corresponding dimensions of the cross-section are typically up to 30 cm. As is well known, the results of bending strength were influenced by the dimensions of the glulam. The European standard for glulam made of softwoods [7] uses beams with a height or depth of 600 mm as the reference. If the overall height or depth is less than 600 mm, as in this study, the experimental results for the bending strength are expected to be up to 10% higher compared to the reference height of 600 mm. It should also be mentioned that in order to standardize glulam, the characteristic bending strength, i.e., the 5th percentile, should be calculated using a sample size according to [29]. The results presented in this paper include bending tests for glulam made from maple, Turkey oak, and European hornbeam. As stated in Section 3.1, the maple did not meet the requirements for the bond shear strength test, so a failure mode at the glue line is possible. The production of glulam from maple would require a gluing process different from that described in this paper. European hornbeam showed high bending strength with planing alone, so further research is necessary to obtain the characteristic properties of glulam from European hornbeam with planed lamellas. Knowledge of the characteristic properties would enable the structural application of these glulam beams, i.e., it could lead to formal technical approval. Likewise, further research is needed to determine the reasons for the increase in stiffness for Turkey oak due to the rougher surface treatment. Such findings would enable better use of the existing hardwood potential.

## 4. Conclusions

This paper presents an experimental investigation of the application of three different hardwood species for the production of glulam: European hornbeam, Turkey oak, and maple. For each species, three different sets of glulam beams were tested depending on the surface treatment of the lamellas. The surface treatments included planing, planing and sanding with grit 60, and planing and sanding with grit 80. The main findings of this study can be summarized as follows:-The relative bond shear strength test for melamine–urea adhesive and glulam from Turkey oak showed satisfactory results regardless of the surface treatment of the lamellas;-The relative bond shear strength tests for melamine–urea adhesive and glulam from European hornbeam showed satisfactory results in planed and fine-sanded lamellas (grit 80) and unsatisfactory results in rough-sanded lamellas (grit 60);-The relative bond shear strength tests for melamine–urea adhesive and glulam from maple showed unsatisfactory results regardless of the surface treatment of the lamellas.-The bending tests of the glulam showed higher bending strength for the glulam made from European hornbeam compared to the glulam from Turkey oak and maple;-The glulam from European hornbeam and maple showed no significant differences in the bending strength for the different surface treatments of the lamellas.-The glulam from Turkey oak with rough sanding of the lamellas (grit 60) showed significantly higher bending strength and stiffness compared to the glulam with planed or fine-sanded lamellas (grit 80).

## Figures and Tables

**Figure 1 materials-16-01843-f001:**
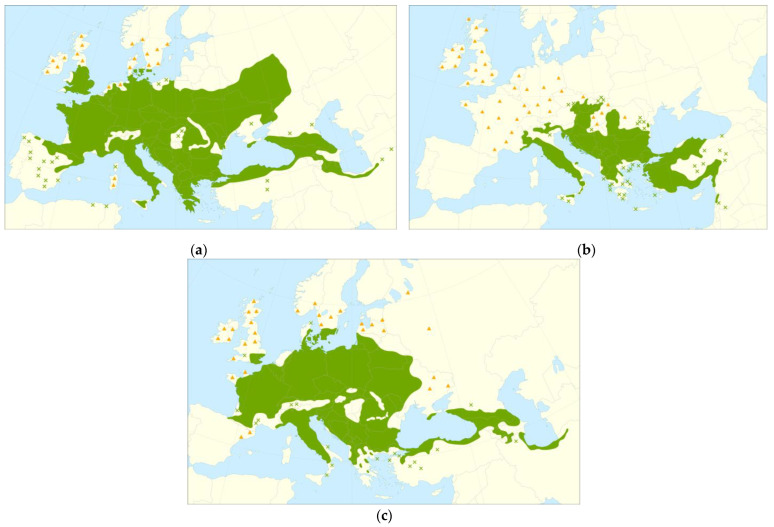
Distribution map for (**a**) maple, (**b**) Turkey oak, and (**c**) European hornbeam according to [24], where the green area presents native range, ✖ is an isolated population, and ▲ is introduced and naturalized.

**Figure 2 materials-16-01843-f002:**
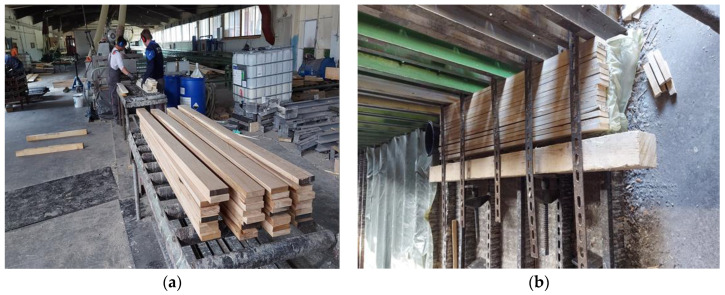
(**a**) Processing laminations on woodworking machine; (**b**) joining laminations together before cramping pressure.

**Figure 3 materials-16-01843-f003:**
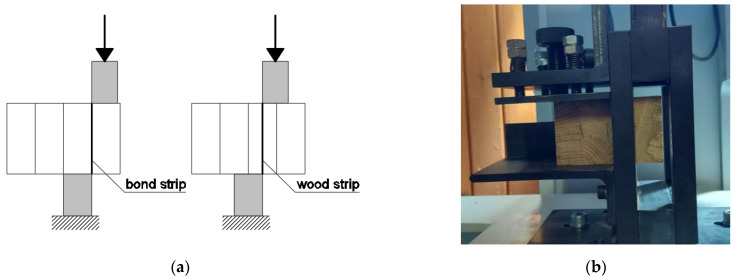
(**a**) Tests for relative bond shear strength; (**b**) testing machine.

**Figure 4 materials-16-01843-f004:**
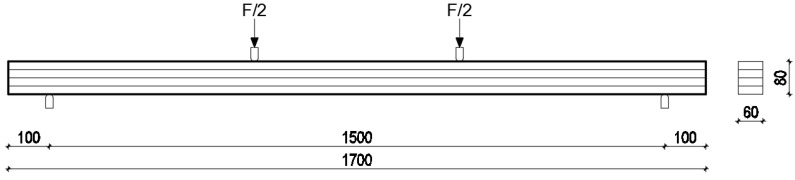
Geometry of the bending test in *mm*.

**Figure 5 materials-16-01843-f005:**
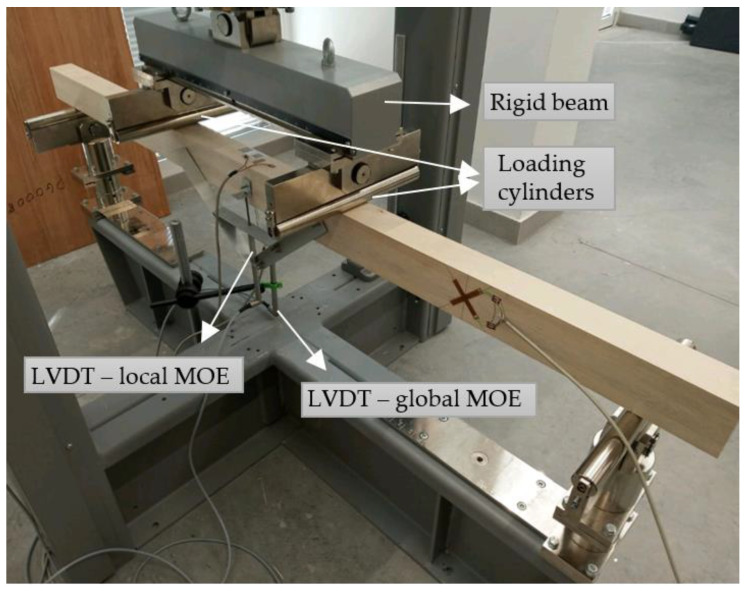
Bending test setup.

**Figure 6 materials-16-01843-f006:**
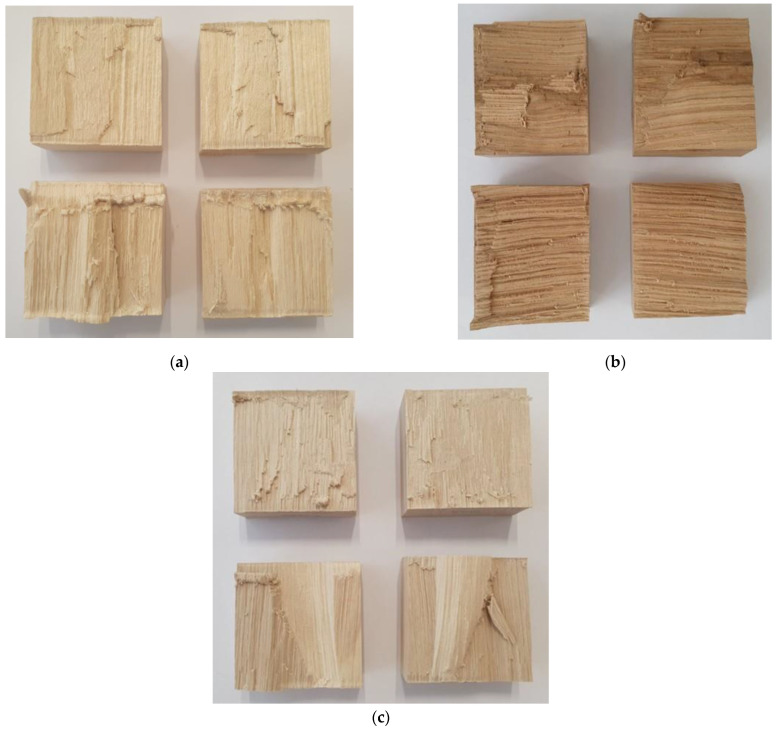
Typical shear failure for (**a**) maple, (**b**) Turkey oak, and (**c**) European hornbeam, with all photos related to planed samples.

**Figure 7 materials-16-01843-f007:**
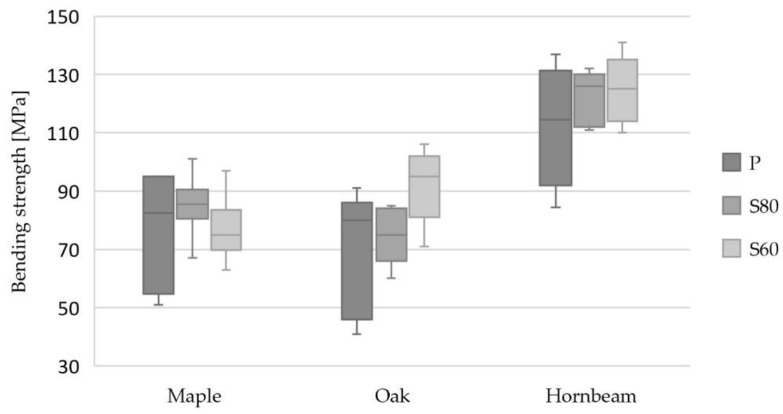
The box plot of the bending strength for glulam beams.

**Figure 8 materials-16-01843-f008:**
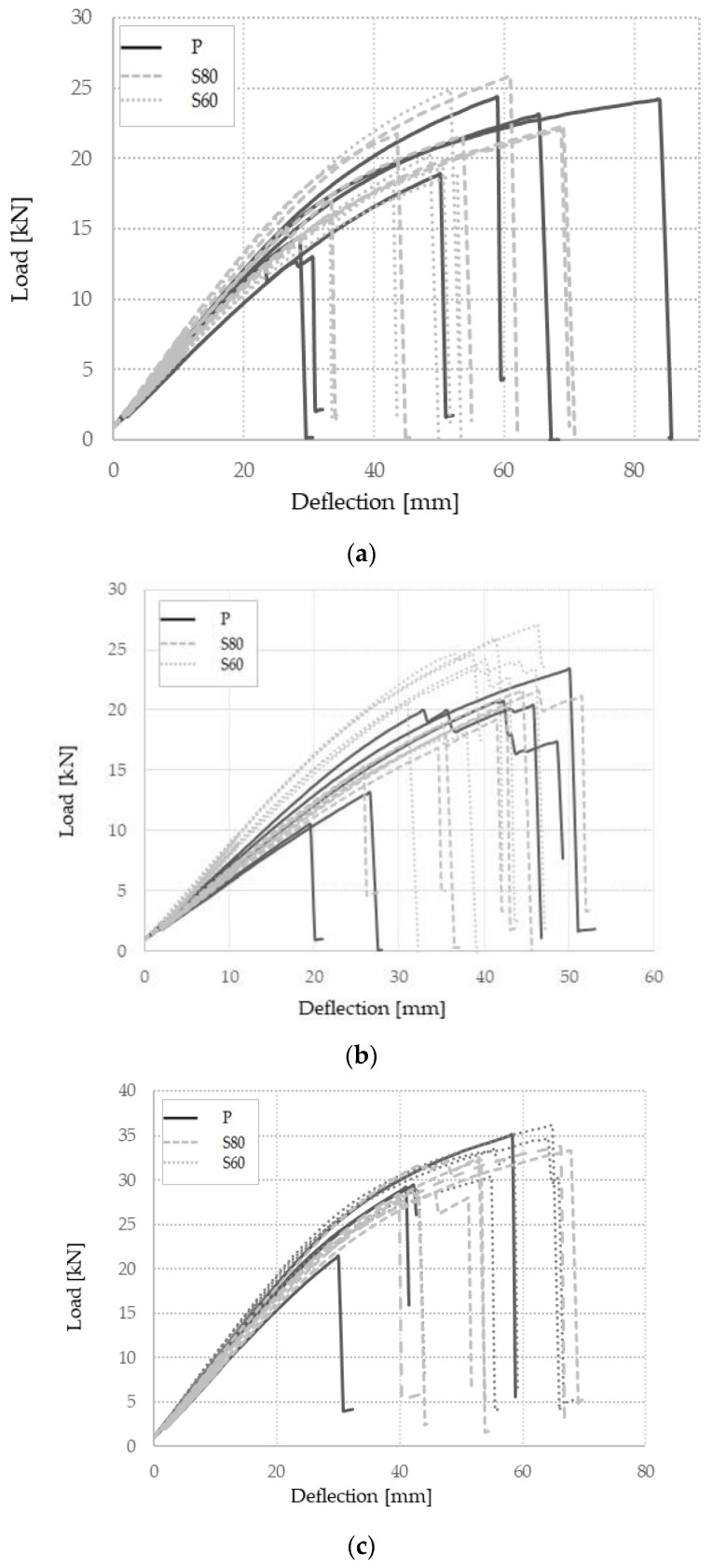
Experimental load–deflection graph for (**a**) maple; (**b**) Turkey oak; (**c**) European hornbeam.

**Table 1 materials-16-01843-t001:** The relative bond shear strength and wood failure percentages of investigated glulam samples (P—planing, S80—fine sanding, S60—rough sanding).

	Maple			Turkey Oak			Hornbeam		
	P	S80	S60	P	S80	S60	P	S80	S60
n	9	9	9	6	6	6	6	6	6
$f_{v,b,mean}$ $\left[\mathrm{Mpa}\right]$	19.60	18.67	18.31	16.32	16.91	16.49	21.63	22.67	19.16
$f_{v,b,mean}$ [Mpa]	21.89	21.89	21.89	14.07	14.07	14.07	22.98	22.98	22.98
$WFP_{mean}$ [%]	81	59	59	100	95	91	95	100	73
$rel\;f_{v,b,mean}$	0.90	0.85	0.84	1.16	1.20	1.17	0.94	0.99	0.83
$f_{v,b,05}$ [Mpa]	16.15	14.40	12.99	12.55	12.18	12.91	14.55	17.99	16.35
$f_{v,w,05}$ [Mpa]	18.70	18.70	18.70	8.61	8.61	8.61	12.12	12.12	12.12
$WFP_{10}$ [%]	15	5	5	100	80	80	90	100	20
$rel\;f_{v,b,05}$	0.86	0.77	0.69	1.46	1.42	1.50	1.20	1.48	1.35

**Table 2 materials-16-01843-t002:** Results of the bending strength and local and global MOE of the glulam beams grouped by hardwood species and surface treatment (P—planing, S80—fine sanding, S60—rough sanding).

	Maple			Turkey Oak			Hornbeam		
	P	S80	S60	P	S80	S60	P	S80	S60
n	6	6	6	5	7	7	4	7	7
$f_{m,g,mean}\;\left[\mathrm{Mpa}\right]$	77	85.2	76.8	68.8	74.1	92.3	112.6	121.4	125
$SD_{f_{m,g}}$ [Mpa]	21.9	11.3	10.9	24.8	9.6	13.1	22.8	9.4	11.4
$E_{mg,mean}$ $\left[\mathrm{Mpa}\right]$	11,350	11,924	11,121	11,836	11,797	15,928	17,580	17,173	18,281
$SD_{E_{m,g}}$ [Mpa]	1066.8	906.2	1029.1	1677.3	607.5	1852.1	1343.6	859.7	1769.3
$E_{m,l,mean}\;\left[\mathrm{Mpa}\right]$	11,547	11,628	11,313	11,803	12,259	16,868	18,686	18,152	19,072
$SD_{E_{m,l}}$ [Mpa]	1078	1257.2	1433.8	1445.1	970	2116.3	936.9	1853.5	1830.6

**Table 3 materials-16-01843-t003:** Results of ANOVA for bending strength of glulam beams made of Turkey oak.

Source of Variation	SS	df	MS	F	*p*-Value	$F_{crit}$
Between groups	1917.5	2	958.7	4.6	0.025	3.633
Within groups	3275.1	16	204.7			
Total	5192.6	18				

**Table 4 materials-16-01843-t004:** Results of DMRT for bending strength of glulam beams made of Turkey oak.

Contrast	Difference	Standardized Difference	Critical Value	Pr > Diff	Significant
S60 ^1^ vs. P ^2^	23.486	2.803	3.144	0.032	Yes
S60 vs. S80 ^3^	18.143	2.372	2.998	0.031	Yes
S80 vs. P	5.343	0.638	2.998	0.533	No

^1^ Rough sanding; ^2^ planing; ^3^ fine sanding.

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
