# Peer review of "An Experimental Investigation of Hardwoods Harvested in Croatian Forests for the Production of Glued Laminated Timber"

_materials, 2023, doi:10.3390/ma16051843_

Round 1
Reviewer 1 Report
Dear Authors,
I have some comments about your work:
1. Let the necessity of research follow from the introduction.
It should be supplemented with details related to forest communities
with selected species (table, map). The whole work should be clearly coherent
2. L 46 "In general, hardwoods have higher densities and 46 tensile strengths, ..." it is not true. There is ca 60 000 wood species in the world. This statment is too general. I suggest to avoid such general simplification.
"Hardwoods are also considered favorable in terms of aesthetics, durability and fire resistance" - it is not true. Think about basewood, poplar and many others and revised that part.
L 72 "The aim of this paper is to present experimental investigations on GLT made from 72 hardwood species which are present in Croatian forests but are not included" Why are they not included and why did you chose those and not other species. There is muc more wood species in Croatia...
You explain that only in case European hornbeam.
L 100 "The measured moisture content of lamellas was between 8 and 15 %" it is a wide range. Give more information of wood origin and its properties (such as density variation, details of moisture content and others). All of that is crucial for properties of finished product.
more, "Specie" is not singular form, it is necessary to check English in all paper.
I do not see any statistcal analysis. Without that no one is able to conlude proper information about results of your work.
Please revise your article in this aspect.
There is also no discussion or comparison with other papers. You are not the first who made such tests. Without the discussion there is a lack of scientific soundness.
Reviewer 2 Report
Please see the attachment. Thanks.

Reviewer 3 Report
The paper focuses on the properties of glued laminated timber made of species that are not listed in the European Assessment Document related to hardwood glulam.
The manuscript is well written, and the clear objectives stated in the introduction were accomplished using standard methods. Besides a few minor problems, I recommend the study for publishing in Materials.
Comments:
The data statistics could be enhanced. In the conclusions, the authors mentioned significant differences in strength (row 319). Authors could test a factor of surface treatment using ANOVA. Duncan’s test could compare mean value differences and confirm significant/nonsignificant differences in the strength/modulus of glulam beams.
Row 71: Is it correct to test wet conditions for service class 1? It is confusing since class 1 defines wood in dry conditions. Please explain or remove it.
Row 104:
For each hardwood specie, (comma) one set …
Figure 4 shows the shear field measurement. I am just wondering; did you get any satisfactory results on the shear modulus? Any comments on this?
Table 1 and Table 2:
The abbreviations (P, S80, S60) should be defined in the title or the footnotes of the table. Please add units in the first column. WFP is in %, contrary to the text description in row 196. Please unify the WFP units.
Figure 7:
The legend (P, S80, S60) should be explained in the title. Please, check the presented data. The maple glulam samples give the highest force values (7a). Assuming the same span and dimensions there is inconsistency with the strength value listed in Table 2 and Figure 6. If Figure 7 is correct, please add a table of the glulam beam dimensions in Materials and Methods.
Round 2
Reviewer 1 Report
Dear Authors,
I see that you use my comments from the report 1 to improve your paper.
Some of them still requires modification. In this form such mistakes that you mised should be in scientific paper in any journal, especially in introduction part. If you want to write about wood, please read more about this material, more taht one or two standards that are made for industry, not for scientis.
Specific comments in attached files.

Reviewer 2 Report
It could be accepted in its present form.
Author Response
Thank you for your time.